# Multi Task Inverse Reinforcement Learning for Common Sense Reward

## Abstract

One of the challenges in applying reinforcement learning in a complex real-world environment lies in providing the agent with a sufficiently detailed reward function. Any misalignment between the reward and the desired behavior can result in unwanted outcomes. This may lead to issues like "reward hacking" where the agent maximizes rewards by unintended behavior. In this work, we propose to disentangle the reward into two distinct parts. A simple task-specific reward, outlining the particulars of the task at hand, and an unknown *common-sense* reward, indicating the expected behavior of the agent within the environment. We then explore how this common-sense reward can be learned from expert demonstrations. We first show that inverse reinforcement learning, even when it succeeds in training an agent, does not learn a useful reward function. That is, training a new agent with the learned reward does not impair the desired behaviors. We then demonstrate that this problem can be solved by training simultaneously on multiple tasks. That is, multi-task inverse reinforcement learning can learn a useful reward function.

## 1 Introduction

Reinforcement Learning (RL) is a machine learning paradigm where an agent learns to make decisions by interacting with an environment (Sutton and Barto, 2018). The agent seeks to maximize a cumulative reward signal received in response to its actions. By maximizing this objective, the agent learns a policy that dictates its behavior within the environment. However, in many real-world applications, one needs to design the reward function so that it precisely defines the target behavior. In these scenarios, designing a suitable reward function becomes a key challenge that practitioners must address in order to train an agent with the desired behavior. This problem was expressed in Dewey (2014) as the *The Reward Engineering Principle:* "As reinforcement-learning-based AI systems become more general and autonomous, the design of reward mechanisms that elicit desired behaviours becomes both more important and more difficult."

Since the agent aims to maximize the reward, any misalignment between the reward and the agent's intended actions can lead to undesirable outcomes. In extreme cases, this misalignment can lead to "reward hacking" Skalse et al. (2022), where the agent successfully maximizes the reward without achieving the desired goal. For instance, in Clark and Amodei (2022), the authors described a scenario where an agent, trained on the CoastRunners boat racing game learned unwanted behavior. The agent repeatedly knocked out targets in an infinite loop to maximize rewards without ever completing the race. Hence, a crucial element in RL is the design of an effective reward function to ensure that agents, trained to maximize this reward, learn the desired behavior, especially when designing agents for operation within complex real-world environments.

To address the reward design problem, we first argue that there is a natural way to split the reward function into two components. One is a task-specific reward that solely defines the goal that the agent aims to accomplish. The second is a task-agnostic reward that describes how the agent should behave in the environment while achieving this goal. We refer to

this task-agnostic reward as the *common-sense* reward, cs-reward for short, as it represents "the basic level of practical knowledge and judgment that we all need to help us live in a reasonable and safe way", using the Cambridge dictionary definition of common sense.

Furthermore, we argue that while the overall reward is complex, in many cases the task-specific part should be relatively simple to design. For example, consider a scenario where a household robot is assigned chores like throwing garbage and mopping floors. Crafting a reward using computer vision models to verify goal completion is not a significant challenge. However, the cs-reward should be much more complex as it needs to account for a wide variety of cases the agent might encounter. Beyond completing specific tasks, the robot must safely navigate spaces, handle delicate objects carefully, conserve electricity, and carry out other actions, such as closing cupboard doors.

Based on this, we argue that disentangling the reward is a natural assumption that can aid the reward design problem. Specifically, we propose to separate the reward function into the task-specific reward, which we assume to be known or learned easily, and the common-sense reward, which is unknown. We then try to learn the shared common-sense reward from expert demonstrations. A natural approach is to use Inverse Reinforcement Learning (IRL), Arora and Doshi (2021) where an agent is trained to imitate the behavior of an expert by simultaneously learning a reward and an agent that tries to maximize said reward. Unfortunately, we show empirically that even when the IRL produces an agent with the desired behavior, it does not learn a meaningful reward. In other words, when attempting to train a new agent from scratch using the learned reward, the desired behavior is not achieved.

An important distinction between this work and most prior works on IRL is that we are interested in the reward function itself, while in most cases the reward serves as a tool to imitate the expert. One intuitive explanation as to why IRL fails to learn a useful reward is its strong connections with the discriminator in Generative Adversarial Networks (GANs) Finn et al. (2016); Ho and Ermon (2016). In the ideal case, the GAN discriminator converges to a non-informative constant function. Our main question is "will a *new* agent trained from scratch with our cs-reward gain the designed behavior?".

We aim to address this challenge by leveraging our previous assumptions about the reward structure. Specifically, we utilize multi-task IRL to learn a task-independent shared cs-reward. We term our approach MT-CSIRL. Intuitively, this allows us to combine information from multiple different experts to avoid learning task-specific behavior and spurious correlations. This directs the learned reward to emphasize the underlying shared common-sense reward. To show the potential of our proposed disentanglement, we designed two simple synthetic common sense rewards on Meta-world benchmark Yu et al. (2020b). We show that even in this simple scenario IRL fails to learn a useful reward and demonstrates the importance of multi-task learning over various tasks to learn a useful and transferable reward.

To conclude, one important contribution of our work is the proposed disentanglement of the reward into the task-specific and task-independent components, formulating the common-sense reward. This formulation has many important advantages. First, we can easily combine information from different tasks, which we show plays a key role in learning useful reward functions. Second, the learned cs-reward can efficiently be transferred to new tasks. Another important contribution is showing empirically that IRL training might fail to learn a proper reward, despite successfully imitating the expert. We then demonstrate how multi-task learning can play an important role in overcoming this difficulty. Our code is available under anonymity at: https://anonymous.4open.science/r/irl-mtl-cs-8572

## 2 Related Work

**Multi-task RL** In multi-task learning, a model is trained to perform multiple tasks simultaneously while leveraging shared representations across tasks to improve overall performance and efficiency Ruder (2017); Caruana (1997). Extensive research has been conducted on multi-task RL and IRL, focusing on utilizing shared properties and structures among tasks Arora et al. (2020); Yang et al. (2020); Vithayathil Varghese and Mahmoud;

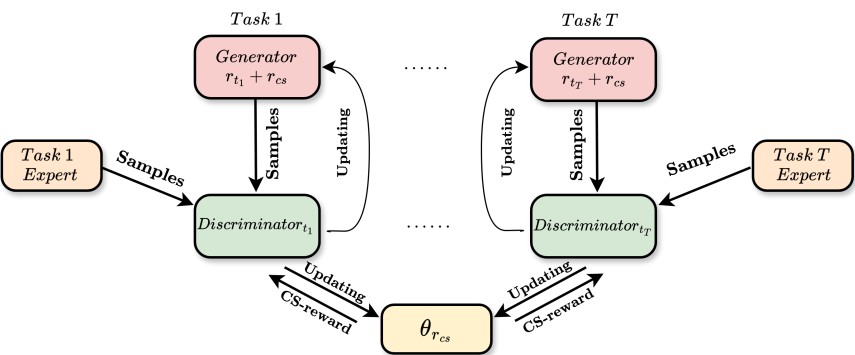

Figure 1: MT-CSIRL architecture overview

Sodhani et al. (2021); Chen et al. (2023); Zhang et al. (2023), overcoming negative transfer Yu et al. (2020a); Liu et al. (2021); Navon et al. (2022) and performing rapid adaptation to novel tasks Yu et al. (2021). MTL was also employed in the IRL setting by learning a policy that imitates a mixture of expert demonstrations. MT-IRL generally focuses on the meta-learning setup. Seyed Ghasemipour et al. (2019); Yu et al. (2019) propose learning a policy conditioned on a trained task context vector. Finn et al. (2017b); Yu et al. (2018) propose a MAML-based Finn et al. (2017a) approach that can adapt a trained policy to a new task with only a few gradient steps. Xu et al. (2019) proposed to learn a prior over reward functions. In Rakelly et al.; Yu et al. (2019) the authors learn a family of rewards by using probabilistic context variables. Differently from these approaches, this paper proposes a method to transfer desired behavior that is not task-specific, without additional adaptation. Another IRL work that separates between task-specific and task agnostic rewards is Chen et al. (2020). In this work, they propose a method to jointly infer a task goal and humans' strategic preferences via network distillation. The main difference between the Reward Network Distillation work and our work is, that in their approach they learn the task-specific reward as the shared reward.

**Regulated and Constrained IRL** One line of research that has strong similarities to this work is inverse constrained learning Malik et al. (2021), and more specifically, multi-task inverse constrained learning Lindner et al. (2023); Kim et al. (2023). In inverse constrained learning the goal is to learn a set of safety constraints from expert demonstrations. While these safety constraints are conceptually similar to our common-sense reward, our common-sense reward is more general. One can consider a hard constraint as a reward that is $-\infty$ for the forbidden set and zero otherwise. This cannot take into account more subtle effects such as a small negative reward for using up a resource, e.g. energy, that we want to discourage the agent from needlessly wasting, but we do not want to stop it from doing so entirely. Another related IRL work is Variational Discriminator Bottleneck Peng et al. (2018). they propose a general technique to constrain information flow in the discriminator by means of an information bottleneck, that can be combined with adversarial inverse reinforcement learning to learn parsimonious reward functions that can be transferred and re-optimized in new settings.

## 3 BACKGROUND

**A Markov Decision Process (MDP).** An MDP is a mathematical framework for modeling sequential decision-making in stochastic environments, central to reinforcement learning. An MDP consists of a tuple $(\mathcal{S}, \mathcal{A}, \mathcal{T}, r, \gamma)$, where $\mathcal{S}$ represents the set of states, $\mathcal{A}$ is the set of actions, $\mathcal{T}$ denotes the state transition probabilities, $r$ is a reward function, and $\gamma$ is the discount factor. Agents in an MDP interact with an environment via a policy $\pi$ that selects actions from $\mathcal{A}$ in a given state $s$ from the distribution $\pi(a|s)$. After action $a$ is selected, the agent transitions to a new state $s'$ and receives a reward $r(s, a, s')$. The state transition probabilities are defined as $\mathcal{T}(s'|s, a)$, representing the probability of transitioning

to state $s'$ given that action $a$ is taken in state $s$. Importantly, the transition and rewards are Markovian, depending solely on the current state and action. The main goal in reinforcement learning is to train an agent to maximize the total future discounted rewards $\mathbb{E}_\pi[\sum_{t=0}^{\infty} \gamma^t r(s_t, a_t, s_{t+1})]$ in an MDP by repeated interactions with the environment.

### 3.1 Inverse Reinforcement Learning

An important RL scenario is imitation learning where the goal is to train an agent on an MDP without having access to the reward, but with expert demonstrations. Commonly, we assume the expert is optimal or near-optimal with regard to the unknown reward. One approach to imitation learning is Inverse Reinforcement Learning (IRL) Arora and Doshi (2018), where we imitate the expert by learning a reward and a policy maximizing it to match the expert demonstrations. In many early IRL approaches, the policy was fully trained at every stage until convergence by using a current reward Abbeel and Ng (2004); Ng and Russell (2000). However, this approach is computationally expensive as it requires training an RL agent from scratch for each reward update. Hence, this approach does not scale well to modern deep reinforcement learning, where the training process can be much more expensive. Current IRL approaches are centered more around training both the reward and the agent simultaneously, updating each one in turn. An important implication of this approach is that it makes the meaning of the learned reward much more uncertain, as there are no guarantees that an agent trained from scratch on the final learned reward will train properly. A visualization of such phenomena is shown in the Experiment section, in Fig. 2. It presents a comparison between training an agent with the learned reward, during IRL process, versus training an agent from scratch using the final learned reward.

**Adversarial Inverse Reinforcement Learning.** The pioneering works Ho and Ermon (2016); Finn et al. (2016) were the first to explore the connection of generative adversarial networks (GANs) to inverse reinforcement learning (IRL), specifically the connection between IRL reward and the GAN discriminator. Their adversarial approach became the primary approach for training IRL systems Fu et al. (2018); Jeon et al. (2021); Han et al. (2022). For simplicity, we will base our work on the Adversarial Inverse Reinforcement Learning (AIRL) Fu et al. (2018) framework, which we found to work well in our experiments.

In AIRL, we train a generator and discriminator simultaneously where the generator is the stochastic policy which we train to fool the discriminator, while the discriminator is trained to distinguish between expert trajectories and generated trajectories. The discriminator in AIRL is formulated as:

$$D(s, a, s') = \frac{\exp f_\theta(s, a, s')}{(\exp f_\theta(s, a, s')) + \pi(a \mid s)}, \tag{1}$$

where $f_\theta(s, a, s')$ encapsulates the learned reward structure, and $\pi(a \mid s)$ is the policy's action probability. This formulation leads to a reward function $r_\theta(s, a, s')$ derived as:

$$r_\theta(s, a, s') = \log D(s, a, s') - \log(1 - D(s, a, s')) \tag{2}$$

$$= f_\theta(s, a, s') - \log \pi(a \mid s). \tag{3}$$

Intuitively, we get a high reward when the discriminator is confident that $(s, a, s')$ belongs to and expert, and a low reward when it is confident $(s, a, s')$ belongs to the agent.

The adversarial loss in AIRL aims to optimize the discriminator to accurately distinguish between expert and agent trajectories. It is defined as:

$$\mathcal{L}(\theta) = -\mathbb{E}_\mathcal{D}[\log D_\theta(s, a, s')] - \mathbb{E}_\pi[\log(1 - D_\theta(s, a, s'))], \tag{4}$$

where $\mathcal{D}$ represents expert demonstrations, and $\pi$ is the policy of the agent.

## 4 Method

### 4.1 Preliminaries

Our method relies on the AIRL framework, leveraging its GAN-IRL architecture to infer a reward that maximizes a general common-sense behavior. Differing from AIRL, our method relies on multi-task learning; therefore, we consider a multi-task IRL setup consisting of a set of tasks $\{t; t = 1...T\}$, where each task is defined by an MDP $(\mathcal{S}, \mathcal{A}, \mathcal{T}, r_t, \gamma)$. Each task $t$, has a set of demonstrations $\mathcal{D}_t$ from an expert's policy $\pi_t^*$. We assume that each expert's policy maximizes a combination of a task-specific reward, $\bar{r}_t$, and a general common sense reward, $r_{cs}$. We aim to recover the task-agnostic common-sense reward, which captures the desired behaviors shared among experts.

### 4.2 Learning the Common-sense Reward

Our goal is to learn the task-agnostic common-sense reward from expert demonstrations by employing Multi-Task Inverse Reinforcement Learning. Our main assumption is that for each task $t$, the loss $r_t(s, a, s')$ has the following structure:

$$r_t(s, a, s') = \bar{r}_t(s, a, s') + r_{cs}(s, a, s'), \tag{5}$$

and that $\bar{r}_t$ is known. This additive split can be modified to other ways to disentangle the rewards, but we use it here for simplicity. This allows for a simple MT-IRL approach where each task policy is trained with a separate weight vector $\pi_{w_t}(a|s)$ and the discriminators for each task share weights, capturing the common-sense reward via:

$$D_\theta^t(s, a, s') = \frac{\exp\left(f_\theta(s, a, s') + \bar{r}_t(s, a, s')\right)}{\exp\left(f_\theta(s, a, s') + \bar{r}_t(s, a, s')\right) + \pi_{w_t}(a|s)}. \tag{6}$$

At each iteration, we pick a task $t$, and update the policy based on the following reward $r_t(s, a, s') = \bar{r}_t(s, a, s') + f_\theta(s, a, s') - \log \pi_t(a \mid s)$, where $f_\theta(\cdot)$ is our learned cs-reward, shared across tasks and designed to capture task-agnostic behaviors.

We then update the task discriminator $D_\theta^t$ with the adversarial loss, similarly to AIRL:

$$\mathcal{L}(\theta) = -\mathbb{E}_{\mathcal{D}_t}[\log D_\theta^t(s, a, s')] - \mathbb{E}_{\pi_{\omega_t}}[\log(1 - D_\theta^t(s, a, s'))], \tag{7}$$

The task discriminators are updated using task demonstrations $\mathcal{D}_t$, and trajectories sampled from the current policy $\tau_t \sim \pi_{\omega_t}$. A full description of our MT-CSIRL method is provided in Alg. 1.

The key aspect of our approach is the disentanglement between task-specific and task-agnostic rewards. This allows us to train on multiple environments while sharing weights among the different discriminators (see Fig. 1). Intuitively, training the shared weights of the different discriminators to simultaneously distinguish expert demonstrations across various environments, avoids learning task-specific behaviors or spurious correlations. We will show empirically in Sec. 5 that training on a variety of tasks is important for learning a transferable common-sense reward function.

### 4.3 Curriculum Learning

We found in our experiments, that training directly with the discriminator in Eq. (6) returns sub-optimal performance on the main task. We hypothesis that, despite having the exact task-specific reward, training with it poses challenges due to the noise introduced by the common-sense reward. This challenge is particularly prominent in the early stages of optimization, given its random initialization.

To overcome this issue and avoid any serious degradation to task-specific performance, we devised a simple curriculum learning approach Bengio et al. (2009). Instead of updating the policy with the combined reward $r_t(s, a, s') = \bar{r}_t(s, a, s') + f_\theta(s, a, s')$, we use $r_t(s, a, s') = \bar{r}_t(s, a, s') + \alpha(\bar{r}_t(s, a, s'))f_\theta(s, a, s')$ with the adaptive weighting $\alpha(\bar{r}_t(s, a, s')) = \frac{\bar{r}_{ave}}{R_{MAX}}$ where

---

**Algorithm 1** MT-CSIRL

---

1: **Input:** Expert demonstrations $\{\mathcal{D}_{t_1}, \ldots, \mathcal{D}_{t_T}\}$, number of iterations $N$, discriminator updates per iteration $D_{\text{updates}}$, number of tasks $T$, task-specific rewards $r_t$, generator updates per iteration $G_{\text{updates}}$
2: **Initialize:** Policy parameters $\omega_t$, discriminator parameters $\theta$
3: **for** $i = 1$ **to** $N$ **do**
4:      **for** $t = 1$ **to** $T$ **do**
5:          **for** $j = 1$ **to** $D_{\text{updates}}$ **do**
6:              Sample trajectories $\tau_t \sim \pi_{\omega_t}$
7:              Update discriminator $\theta$ using gradient ascent
8:          **end for**
9:          **for** $k = 1$ **to** $G_{\text{updates}}$ **do**
10:             Sample trajectories $\tau_t \sim \pi_{\omega_t}$
11:             Update $\omega_t$ using $\bar{r}_t + \alpha \cdot r_{\text{CS}}$
12:          **end for**
13:      **end for**
14: **end for**

---

$\bar{r}_t(s, a, s') \in [0, R_{MAX}]$, and $\bar{r}_{ave}$ is the historical average of task-specific rewards over a window of 256 steps, for stability. This allows us to gradually increase the weighting of our learned reward as the policy improves on the main task. This way the common-sense reward does not have a significant effect at the beginning of training, and thus only impacts the policy after it had several update rounds. We note that this does not affect the discriminator update, which still uses Eq. 6. Also, when training a new agent with our learned reward, we use the standard aggregation $r_t(s, a, s') = \bar{r}_t(s, a, s') + f_\theta(s, a, s')$.

### 4.4 EXTENSION TO UNKNOWN TASK REWARDS

So far, we assumed that the task reward is known and focused on the task-agnostic reward. While we believe this is an important and common scenario, we will show our approach is not limited to this setting. Here, we extend our approach to the case where we have expert demonstrations from $T$ tasks, however the task-specific rewards are unknown. As before, we assume each expert maximizes a combination of the task-specific and common-sense behavior rewards, and we aim to recover a common-sense reward, $r_{cs}$, which will capture the shared desired behavior.

In these cases, we need to simultaneously learn per-task reward functions from each expert's demonstrations and a shared common sense reward from all experts. For each expert, we train a task discriminator to learn the task-specific reward:

$$D_{\phi_t}(s, a, s') = \frac{\exp\left(\bar{r}_{\phi_t}(s, a, s')\right)}{\exp\left(\bar{r}_{\phi_t}(s, a, s')\right) + \pi_{w_t}(a|s)}. \tag{8}$$

where $\phi_t$ are the learned parameters for the reward of task $t$. In addition, we train a shared discriminator for the common sense reward:

$$D_\theta(s, a, s') = \frac{\exp\left(f_\theta(s, a, s')\right)}{\exp\left(f_\theta(s, a, s')\right) + \pi_{w_t}(a|s)}. \tag{9}$$

We then use each task reward and the shared common sense reward to update each task generator policy. This process of our extension for multi learned task (MT-CSIRL+LT) is depicted in Appendix A.4

## 5 EXPERIMENTS

For our experiments, we use the Meta-world benchmark Yu et al. (2020b). It provides a diverse set of robotic manipulation tasks, which share the same robot, action space, and

observation space. Thus, this benchmark is suitable for evaluating the effectiveness and transferability of our inverse reinforcement learning method across various scenarios. We trained all policies using the Soft Actor-Critic (SAC) algorithm Haarnoja et al. (2018). In order to emphasize our primary goal of learning transferable rewards, we limit our experimental setup to Meta-world tasks on which SAC performed well, according to the Meta-world's paper. We used the exact training process and SAC hyperparameters as in Yu et al. (2020b). We repeat the experiments five times using different random initializations and report the mean and standard deviation of the performance. Using the Meta-world terminology, there are several distinct tasks, e.g., reach-wall and drawer-close. For each task, there are several variations with distinct targets. For example, in the drawer-close task, a different target would be a different location of the drawer. We also note that Meta-world has two versions for each task. All experiments in this paper use the second version, i.e., reach-wall is reach-wall-V2, and we omit the V2 for brevity.

The goal of our experiments is to demonstrate that IRL fails to learn a useful cs-reward and that training the cs-reward on multiple tasks enables us to learn such a reward. To show that, we perform several experiments, each time learning the cs-reward (which we will describe shortly) from a more diverse set of tasks. See Fig. 8 in Appendix B for a visual summary of our experiments.

**Common-Sense Rewards:** As Meta-world only contains task-specific rewards, we designed two simple common-sense rewards for our experiments. The explicit common-sense reward is not given directly to the agent during its IRL training process; instead, it was learned through expert demonstrations. The common-sense reward is used to score the agent, for evaluation purposes.

Our first cs-reward directs the agent to move one of the key points along the robotic arm (whose 3D location is part of the observation vector) with a target velocity $v_{target}$. The reward function is given by

$$r_{CS}(s, a, s') = -C_v \cdot |\|\ell(s') - \ell(s))\|_2 - v_{target}|,\tag{10}$$

where $\ell(s)$ is the 3D location of the selected key point given by the observation vector. The second reward directs the $L_1$ norm of the action towards a target value $n_{target}$. The reward is defined as follows,

$$r_{CS}(s, a, s') = -C_n \cdot |\|a\|_1 - n_{target}|.\tag{11}$$

We name these reward functions the *velocity* and *action norm* cs-rewards, respectively. See Appendix B for further details. One important property of these rewards is that they do not depend on the task or environment and, as such, should be easily transferable. We specifically designed these simple, common-sense behaviors to show how IRL struggles to learn a transferable reward even in this simple setting.

**Baselines:** In our experiments, we evaluate and compare the following methods: (1) *Expert*: RL trained with the task and ground-truth cs-reward; (2) *SAC*: An RL trained with only the task reward; (3) *MT-AIRL* Fu et al. (2018), Implementation of multi-task AIRL for learning the entire combined reward per task; (4) *MT-VAIRL* and (5) *MT-VAIRL-GP* Peng et al. (2018), implementation of multi-task VAIRL and multi-task VAIRL-GP for learning the entire combined reward per task. Our methods: (6) *MT-CSIRL*, (Alg. 1) and (7) *MT-CSIRL+LT* (Alg. 2). Since we introduce a novel setting, none of the existing standard solutions we compare to use the separation between task-agnostic reward and task-specific reward. However, they still allow us to infer the benefits of utilizing this assumption.

## 5.1 Learning CS-Reward from a Single Task

We will show empirically that, even for simple common-sense rewards for which the IRL process successfully trains an agent, the final reward might not be useful for training a new agent from scratch. This issue occurs even when applying the reward on the exact same task and target. To show this, we train an expert with the task-specific (i.e., reach-wall) and ground truth common-sense reward for each of our common-sense rewards. In both velocity

and action norm cases, we successfully train experts who maximize both the task rewards and the common-sense rewards. Then, for each expert, we sample a set of trajectories for IRL training. The cs-reward is learned using Alg. 1 with a single task (i.e., $T = 1$). Finally, we train a new agent on the exact same task and target using our learned cs-reward and known task reward. In all of the following experiments in this section, all trained agents achieved 100% success rate on the main task. Therefore, we only report the results on the common-sense rewards.

We present our results in Fig. 2 (a) & (b). During the IRL process, the trained agent effectively acquired the desired common-sense behavior, marked in the horizontal red line. However, when we train a new RL agent with our learned reward (and the task-specific reward) it is indistinguishable from the baseline SAC trained with only the task-specific reward. Similar results with different tasks are shown in Appendix A.1. We note that this phenomenon has been also previously observed in other bi-level optimization scenarios Navon et al. (2020); Vicol et al. (2022).

In Fig. 2 (c), we also present a scatter plot of the learned cs-reward versus the ground-truth cs-reward. This shows a very small correlation between the learned reward and the ground-truth reward (correlation coefficient of 0.18), which further demonstrates that the IRL fails to capture the desired reward.

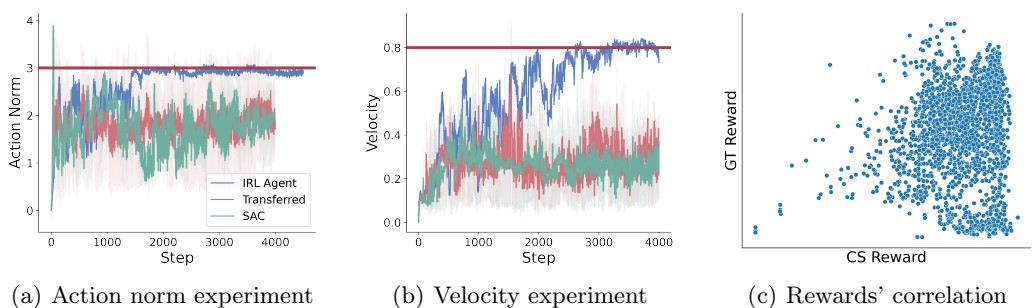

(a) Action norm experiment      (b) Velocity experiment      (c) Rewards' correlation

Figure 2: *Single task cs-reward:* In (a), (b), we plot the rewards during the IRL process (IRL Agent), an RL agent with the learned cs-reward from the IRL process (Transferred), and a baseline RL agent without any common sense component (SAC). The red horizontal line represents the target value in velocity/action norm see Eq. 10 and 11.The scatter plot (c) shows the correlation between the ground-truth reward and the learned CS-Reward.

## 5.2 Learning CS-Reward from Multiple Targets

In the next experiment we show that when we train our IRL on expert demonstration from different targets, we manage to learn a reward that can be used to train new agents on unseen targets. However, it does not transfer to novel tasks. Again, all agents trained in this section achieve 100% success rate so we only show results for the cs-reward. To do that, we train our MT-CSIRL method with experts trained on different targets for the same task, for each of our cs-rewards. We then test how these learned rewards can be used to train on new targets in this environment and how they transfer to other tasks. We execute this experiment on five different Meta-world tasks: reach-wall, button-press-topdown-wall (BP Topdown-wall), drawer-close, push-back and coffee-button. For each task, we train multiple experts, one for each target. We sample trajectories from the experts' policies, and we use this trajectories as the expert demonstrations. We then learn a cs-reward using our method MT-CSIRL. We evaluate the performance of agents trained from scratch using this learned cs-reward on the training task as well as on new unseen tasks. For more experiment details see Appendix A.2.

The results for action norm reward are presented in Table 1, and velocity results are in Appendix A.2. The numbers represent the ratio between the agents' action norm and its target value, i.e., the closer to one, the better. On the diagonal, we see the results on

different targets for the original task, with the baseline results for agents trained only on the task reward in parenthesis. On the off-diagonal, we see the results when transferring to new tasks. When looking at the diagonal elements of Tables 1, we see that our reward transfers well to new targets for the seen task. While our agents do not reach the target values, they still show significant improvement over the baseline. However, the drop in performance on the off-diagonal elements indicates that the learned reward does not transfer well to novel tasks. This is especially interesting when we consider our simple cs-rewards, which do not depend on the specifics of the environment (and independent of the state for action norm).

|  | reach-wall | BP Topdown-wall | drawer-close | push-back | coffee-button |
|---|---|---|---|---|---|
| reach-wall | **0.91**(0.57) | 0.43 | 0.60 | 0.54 | 0.58 |
| BP Topdown-wall | 0.60 | **0.91**(0.61) | 0.58 | 0.60 | 0.59 |
| drawer-close | 0.59 | 0.53 | **0.90**(0.56) | 0.63 | 0.57 |
| push-back | 0.56 | 0.56 | 0.51 | **0.90**(0.62) | 0.55 |
| coffee-button | 0.58 | 0.57 | 0.54 | 0.58 | **0.92**(0.48) |

Table 1: Action Norm cs-reward. Each row represents a different task for MT-CSIRL training, each column represents a task for RL training with learned reward and evaluation. The Numbers represent the ratio between the agents' action norm and its target value. In the diagonal, Action Norm reward for training solely on the task reward appears in parentheses.

### 5.3 Learning CS-Reward from Multiple Tasks

Here we show that when we train the IRL process on experts' trajectories from multiple tasks, we learn a useful cs-reward that can be transferred to new unseen tasks. In order to do that, we perform our MT-CSIRL method again, but this time we train each expert on a different meta-world task.

The results for velocity as cs-reward, and action norm as cs-reward are presented in table 2. As can be easily observed, the MT-CSIRL results show that our learned cs-reward manages to transfer the desired behavior even to unseen tasks, albeit not as strongly as the ground-truth reward. We included the MT-AIRL baseline, which unsurprisingly does not work well, to show the importance of our split between task-specific and task-independent rewards. Without this distinction combining different tasks is non-trivial and can easily harm performance as we see here. We also note that the MT-AIRL baseline shares similarities with Gleave and Habryka (2018), however, they train with a meta-learning approach similar to MAML.

To further illustrate our results we show in Fig. 3 the ground truth cs-reward (both for velocity and action norm) on the unseen test task. The figures show the positive impact of our cs-reward, making the agent's behavior significantly more aligned with the expert. Finally, we show in Fig. 3 a scatter plot of the ground-truth cs-reward versus our learned reward on a new unseen task. As one can see, these rewards are highly correlated (correlation coefficient of 0.88) which again shows that our agent managed to learn the desired reward function. The correlation results in Fig. 3 are for the velocity experiment, the action norm scatter plot is in Appendix A.3.

### 5.4 Learning CS-Reward and Task-Reward from Multiple Tasks

In this section, we implement the extension to our methodology as detailed in 4.4, where we will assume we have expert demonstrations from T tasks but the task-specific rewards are unknown. We train an IRL process on multiple tasks using MT-CSIRL+LT method. Training this process gives as a learned cs-reward, and a learned task-specific reward for each task in the training process. In order to show that the learned cs-reward can be transferred to novel tasks, we train new RL agents with the ground truth task reward and our learned cs-reward (results in Table 2). The difference here is that during the cs-reward training we did not have access to the task rewards. In Appendix A.4 we show how this method performs on novel tasks without the ground truth reward but with expert demonstrations.

| | ACTION NORM | | VELOCITY | |
| AGENT | AVG ACTION NORM | SUCCESS-RATE | AVG VELOCITY | SUCCESS-RATE |
|---|---|---|---|---|
| EXPERT | $2.95 \pm 0.01$ | $100 \pm 0.00$ | $0.81 \pm 0.02$ | $100 \pm 0.00$ |
| SAC | $1.57 \pm 0.48$ | $100 \pm 0.00$ | $0.32 \pm 0.16$ | $100 \pm 0.00$ |
| MT-AIRL | $1.30 \pm 0.47$ | $63.5 \pm 1.20$ | $0.11 \pm 0.01$ | $67.5 \pm 5.90$ |
| MT-VAIRL | $2.29 \pm 0.07$ | $79.6 \pm 4.45$ | $0.38 \pm 0.06$ | $73.3 \pm 2.51$ |
| MT-VAIRL-GP | $2.32 \pm 0.19$ | $75.0 \pm 1.37$ | $0.39 \pm 0.02$ | $76.8 \pm 2.65$ |
| MT-CSIRL (OURS) | $\mathbf{2.75 \pm 0.18}$ | $99.9 \pm 0.01$ | $\mathbf{0.73 \pm 0.07}$ | $99.2 \pm 0.30$ |
| MT-CSIRL+LT (OURS) | $2.70 \pm 0.35$ | $97.9 \pm 0.08$ | $0.70 \pm 0.04$ | $97.2 \pm 0.06$ |

Table 2: Experiment results on unseen tasks with action norm cs-reward and velocity cs-reward, both learned in a multi task learning framework. first with a given task reward during IRL (MT-CSIRL), and second with a learned task reward during IRL (MT-CSIRL+LT).

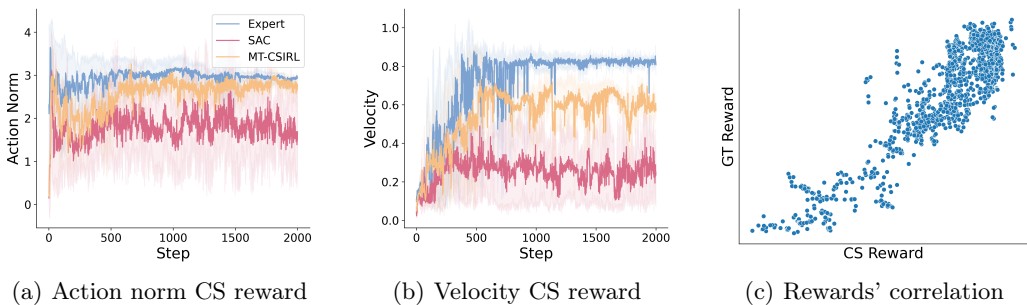

(a) Action norm CS reward    (b) Velocity CS reward    (c) Rewards' correlation

Figure 3: *Ground Truth CS-Reward*: In (a) and (b), We show the ground-truth common sense behavior on three different experiments. Expert: maximizes task reward and ground-truth cs-reward. MT-CSIRL: trained with the learned cs-reward and with the task reward. SAC: "vanilla" training, trained only with task reward. In (c) we visualized the scatter plot between Ground-Truth & Learned CS-Reward from MT-CSIRL method, on Velocity Experiment.

## 6    CONCLUSIONS

In this work, we addressed the important question of how to learn rewards capable of guiding reinforcement learning agents to exhibit desired behavior within our environment. An important step in achieving this goal is to disentangle the task-independent reward, which we name common-sense reward from the task reward. We believe this simple observation can have an important impact as it allows us to focus on learning *how* the agent should behave and not on *what* the agent should be doing. Furthermore, our experiments show that our framework allows us to easily combine expert observations from different tasks and that learning from a variety of different tasks is a key to learning meaningful reward functions. Finally, we observe that current IRL methods, while successful in imitation learning, still struggle to learn a proper reward function. We advocate for further work in this direction, as the reward function can be more than an auxiliary for imitation learning.

## 7    LIMITATION AND BROADER IMPACT

In this work, we experiment with simple synthetic common-sense rewards. These rewards are informative for this study, as they show how current methods do not learn a useful or transferable reward even in this basic setting. However, further research is required with a more realistic common-sense reward. This will require designing a novel and complex RL benchmark and is beyond the scope of this work. Regarding broader impact, this work impacts the way we train agents with better alignment with desired behavior.

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

## A    ADDITIONAL EXPERIMENT RESULTS

This section includes additional experiments' results, introduced in the order presented in the Experiment section 5 of the paper.

### A.1    ADDITIONAL EXPERIMENTAL RESULTS FOR CS-REWARD TRAINED ON A SINGLE TASK

In section 5.1 we explained about figure 2, showing that during the IRL process the trained agent successfully learned the desired common-sense behavior. However, a new agent training on the learned reward does not achieve the same desired behavior. Here we will show those results on two more tasks. In the paper, figure 2 shows graphs for the "reach-wall-v2" task. Results fot "drawer-close-v2", and "button-press-topdown-wall-v2" are shown in Fig. 4 and 5 respectively.

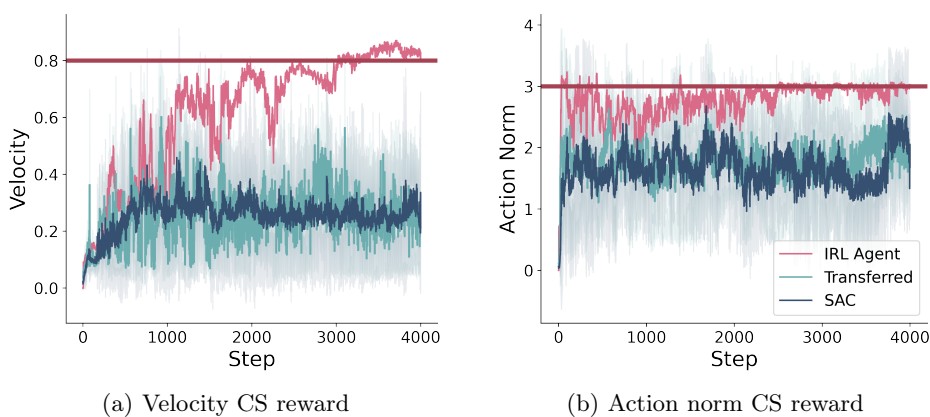

(a) Velocity CS reward                         (b) Action norm CS reward

Figure 4: *Single task cs-reward*: Visualization of the rewards during the IRL process (IRL Agent), an RL agent with the learned cs-reward from the IRL process (Transferred), and a baseline RL agent without any common sense component (SAC). The red horizontal line represents the target value in the ground truth reward, see Eq. 10 and 11. This experiment was conducted on the button-press-topdown-wall setup task.

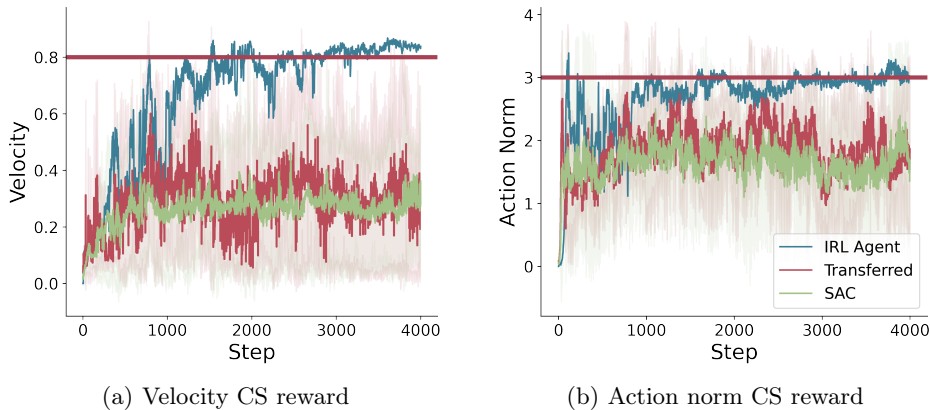

(a) Velocity CS reward                         (b) Action norm CS reward

Figure 5: *Single task cs-reward*:Visualization of the rewards during the IRL process (IRL Agent), an RL agent with the learned cs-reward from the IRL process (Transferred), and a baseline RL agent without any common sense component (SAC). The red horizontal line represents the target value in the ground truth reward, see Eq. 10 and 11. This experiment was conducted on the coffee-button setup task.

In Fig. 2 (c), we present a scatter plot of the learned cs-reward versus the ground-truth cs-reward, which is a result of the experiment in section 5.1. The correlation results in Fig. 2 are for the velocity experiment.

Here we show the correlation between Ground-Truth & Learned CS-Reward Trained on a Single Task, for the action norm experiment. with the scatter plot visualization in Fig. 6. Similarly to the velocity experiment, the action norm experiment shows no correlation between the learned reward and the ground-truth reward, which further demonstrates that the IRL fails to capture the desired reward.

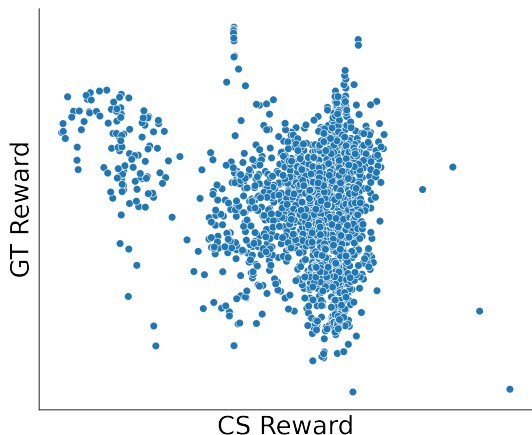

Figure 6: Action Norm Scatter plot of GT & CS-Reward Learned using Single Task

## A.2 Additional experimental results for Learning CS-Reward with Multiple Targets

In section 5.2, we show that training our IRL method on expert demonstrations from different targets allows learning a reward that can train new agents on unseen targets, but it does not transfer to novel tasks. All agents achieve a 100% success rate, so we only show results for the cs-reward.

Here we show, in Table 3, the results of the velocity common-sense reward in the multiple target setup. We train our MT-CSIRL method with experts on different targets for the same task and test these learned rewards on new targets and tasks.

The diagonal elements show our reward transfers well to new targets for the seen task, with the baseline results for agents trained only on the task reward in parenthesis. On the off-diagonal elements indicate poor transfer to novel tasks, highlighting the limitations of our cs-rewards.

As an additional baseline for the multiple targets experiment, we report the velocity when the task is learned using only the task-specific reward, without the learned velocity cs-reward, with the results presented in parentheses.

|  | reach-wall | BP Topdown-wall | drawer-close | push-back | coffee-button |
|---|---|---|---|---|---|
| reach-wall | **0.93**(0.37) | 0.40 | 0.43 | 0.40 | 0.25 |
| BP Topdown-wall | 0.36 | **0.82**(0.24) | 0.50 | 0.31 | 0.21 |
| drawer-close | 0.34 | 0.48 | **0.89**(0.26) | 0.35 | 0.24 |
| push-back | 0.25 | 0.25 | 0.20 | **0.88**(0.28) | 0.22 |
| coffee-button | 0.28 | 0.32 | 0.35 | 0.38 | **0.91**(0.26) |

Table 3: Velocity cs-reward. Each row represents a different task for MT-CSIRL training, each column represents a task for RL training and evaluation. The Numbers represent the ratio between the agents' velocity and its target value, closer to one is better. In the diagonal, Velocity reward for training solely on the task reward appears in parentheses

### A.3 Additional experimental results for Learning CS-Reward with Multiple Tasks

In section 5.3, we show that training our IRL method process on experts' trajectories from multiple tasks allows us to learn a cs-reward that can transfer to new, unseen tasks.

We perform our MT-CSIRL method again, training each expert on a different Meta-world task. In Fig. 7 we visualize a scatter plot of the Ground Truth cs-reward versus our learned reward on a new unseen task for the Action Norm case.

The scatter plot shows a high correlation with a correlation coefficient of 0.86. This indicates that in the velocity case too, our agent successfully learned the desired reward function.

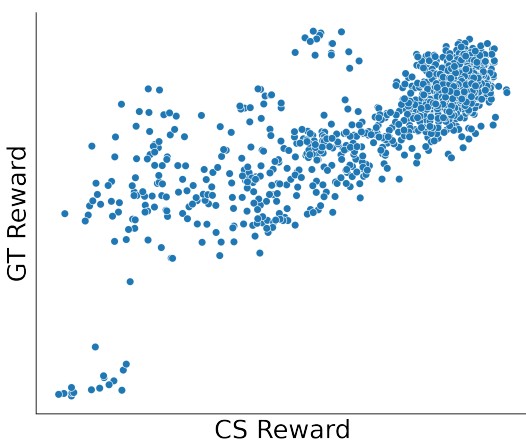

Figure 7: Action Norm Scatter plot of GT & CS-Reward Learned using MT-CSIRL

### A.4 Additional experimental results for Learning CS-Reward and Task-Reward with Multiple Tasks

In section 4.4, an explanation of the extension to unknown task reward method is explained, and in section 5.4, we implement the extension to this method. We assume expert demonstrations from T tasks with unknown task-specific rewards. Using MT-CSIRL+LT, we train the IRL process on multiple tasks, resulting in a learned cs-reward and task-specific rewards for each task.

In this setup, for each task $t$ we have expert demonstration, and we do not have the explicit task reward the expert's demonstrations maximizing. To learn both cs-reward, and task-specific reward, we simultaneously learn per-task reward functions from each expert's demonstrations and a shared common sense reward from all experts. For each expert, we train a task discriminator to learn the task-specific reward.

We use the learned cs-reward and the learned task-specific reward, and we train from scratch an agent that needs to maximize both cs-reward and task-specific reward. Results for this evaluation are shown in Table 4

| ENV | ACTION NORM | | VELOCITY | |
| --- | --- | --- | --- | --- |
| | ACTION NORM RATIO | SUCCESS-RATE | VELOCITY RATIO | SUCCESS-RATE |
| WINDOW-OPEN | 0.88 | $87.80 \pm 1.77$ | 0.90 | $88.7 \pm 1.72$ |
| REACH | 0.91 | $91.25 \pm 1.67$ | 0.89 | $84.2 \pm 2.14$ |
| PLATE-SLIDE | 0.90 | $90.45 \pm 3.01$ | 0.88 | $83.3 \pm 2.32$ |
| FAUCET-OPEN | 0.89 | $88.69 \pm 2.42$ | 0.88 | $92.3 \pm 2.33$ |
| COFFEE-BUTTON | 0.91 | $91.81 \pm 2.71$ | 0.90 | $87.5 \pm 1.23$ |
| DOOR-CLOSE | 0.90 | $86.92 \pm 2.60$ | 0.89 | $90.0 \pm 1.41$ |

Table 4: SAC Agent performance, with Learned Common Sense Reward and Learned Task Reward using MT-CSIRL+LT process. No transformation - learning from scratch the task whose rollouts are seen in IRL process

---

**Algorithm 2** MT-CSIRL+LT

---

1: **Input:** Expert demonstrations $\{\mathcal{D}_{t_1}, \ldots, \mathcal{D}_{t_T}\}$, number of iterations $N$, discriminator updates per iteration $D_{\text{updates}}$, number of tasks $T$, generator updates per iteration $G_{\text{updates}}$
2: **Initialize:** Policy parameters $\omega_t$, common sence discriminator parameters $\theta_{cs}$, and task discriminator parameters $\phi_{t_i}$
3: **for** $i = 1$ **to** $N$ **do**
4:    **for** $t = 1$ **to** $T$ **do**
5:       **for** $j = 1$ **to** $D_{\text{updates}}$ **do**
6:          Sample trajectories $\tau_t \sim \pi_{\omega_t}$
7:          Update CS-discriminator parameters $\theta_{cs}$ using gradient ascent
8:          Update Task-specific discriminator parameters $\phi_{t_i}$ using gradient ascent
9:       **end for**
10:       **for** $k = 1$ **to** $G_{\text{updates}}$ **do**
11:          Sample trajectories $\tau_t \sim \pi_{\omega_t}$
12:          Update $\omega_t$ using $\bar{r}_{t_i} + \alpha \cdot r_{\text{CS}}$
13:       **end for**
14:    **end for**
15: **end for**

---

## B    EXPERIMENTAL DETAILS

**Environments** We evaluate our experiments on the Meta-world benchmark. All tasks are performed by a simulated Sawyer robot. The action space consists of the 3D change of the end-effector and the normalized torque of the gripper fingers, ranging from -1 to 1. The robot either manipulates one object with a variable goal or two objects with a fixed goal. The observation space is a 39-dimensional 6-tuple of the 3D positions of the end-effector, gripper openness, positions and quaternions of two objects, and the goal. If no second object or goal, corresponding values are zeroed out.

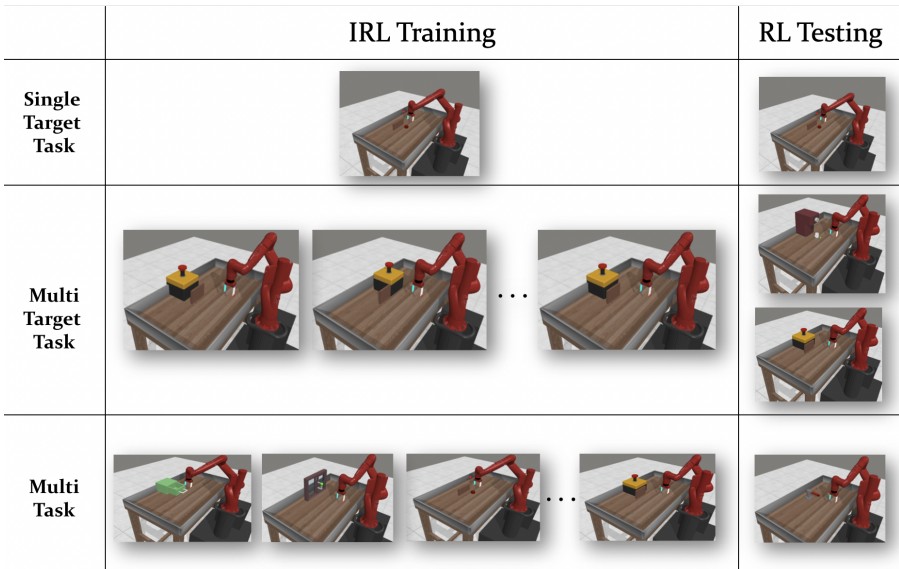

Figure 8: Visualization of train and test tasks for our three experimental setups: Single Target, experiments in section 5.1, experiments in section Multi Target 5.2, and Multi Task, experiments in sections 5.3, 5.4

**Common Sense Reward - Technical Details** In our experimental framework, we utilize the Meta-World benchmark, which incorporates the Sawyer robot, a collaborative robotic arm known for its high precision and 7-degree-of-freedom.

This setup enables us to simulate and evaluate complex tasks, and to define a common sense, which we showed in our paper how we learned and transfer it between tasks.

Our velocity ground truth reward was calculated on the "end effector" part of the Sawyer robot shown in Fig. 9, and the end effector marked with an arrow. The end effector located at the end of the arm, there is an end effector, which can be equipped with different tools or grippers, depending on the task.

In simulation environments like Meta-World, the end effector is often a gripper used for tasks like picking and placing objects.

The location of the end effector is given us by both Mujoco engine and Meta-worlds observation space. In each step we have an access to the end effector's location of states $s$ and $s'$, and we use those locations to calculate the velocity, as explained in section 5.

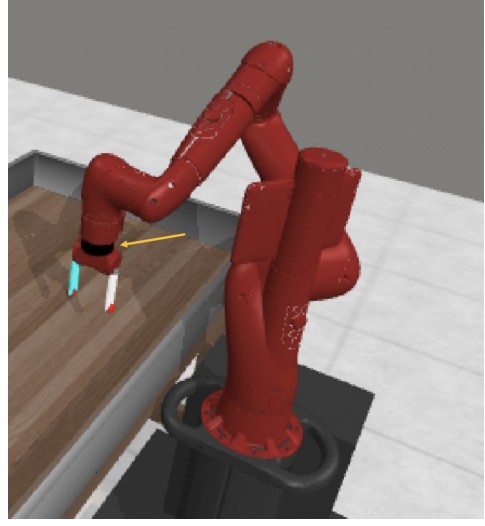

Figure 9: *Sawyer Robot*: The end effector, on which we calculated the velocity, is marked with an arrow.

We introduce two common sense behaviors, as shown in section 5. In the velocity case, we calculate the velocity of the Sawyer's end effector. Our reward function is maximized when the end effector velocity get closer to $C_v$. In our experiments, we set $C_v$ to be 0.8. As for the action norm, our ground truth reward is maximized when the $l_1$ of the action space, gets closer to $C_n$. In our experiments, we set $C_n$ to be 3. It is important to mention that action space is the same for all meta worlds tasks.

**Expert Demonstrations**. In all our experiments, we used experts' demonstration to train the discriminators. To create the expert demonstrations, we run SAC, and train in to maximize two reward functions. The first is the task-specific reward, and the second in the ground truth cs-rewards, as shown in Eq. 10 and 11. For SAC training, our hyperparameters (hp) are similar to the hp detailed in Meta-world benchmark. For hyperparameters values, see Table 5.

**Code Base** We use the garage toolkit, garage contributors (2019) which is a toolkit for developing and evaluating reinforcement learning algorithms, with a library of state-of-the-art implementations. We also utilized the Human-Compatible AI (HCAI) group's project "Imitation Learning Baseline Implementations" Gleave et al. (2022) which provides clean implementations of imitation and reward learning algorithms.

**Experimental Details for CS-Reward Trained on a Single Task.** The cs-reward is learned using Alg. 1, and we run in on the meta-worls envs: reach-wall, button-press-topdown-wall, drawer-close, push-back and coffee-button, separately. We train this experiment with $T = 1$. another input is the expert demonstrations. We train the experts for this experiment as explained above in the current section. For each task, we sampled 60K trajectories. The Discriminator updates per iteration is 15, and the generator updates per iteration is 15. Our generator agent's policy, and the new agent we trained from scratch with the learned cs-reward, are optimized with SAC algorithm. In Fig. 2, 4 and 5, Transferred graph and the SAC graph are averaged across five seeds.

**Experimental Details for Learning CS-Reward with Multiple Targets.** Experimental details are similar to experimental Details for CS-Reward Trained on a Single Task, except here we train Alg. 1 on multi-targets for each task. For each task, we train Alg. 1 on 5 different targets, means $T = 5$. We learned a new cs-reward, and use it to train from scratch a new SAC agent on the same task, but with target. For each task, we train the SAC

| Description | Value |
|---|---|
| **Normal Hyperparameters** | |
| Batch size | 500 |
| Number of epochs | 500 |
| Path length per roll-out | 500 |
| Discount factor | 0.99 |
| **Algorithm-Specific Hyperparameters** | |
| Policy hidden sizes | (256, 256) |
| Activation function of hidden layers | ReLU |
| Policy learning rate | $3 \times 10^{-4}$ |
| Q-function learning rate | $3 \times 10^{-4}$ |
| Policy minimum standard deviation | $e^{-20}$ |
| Policy maximum standard deviation | $e^2$ |
| Gradient steps per epoch | 500 |
| Soft target interpolation parameter | $5 \times 10^{-3}$ |
| Use automatic entropy tuning | True |

Table 5: Hyperparameters used for Garage experiments with Single Task SAC

agent from scratch with the learned cs-reward on five different seeds. The results presented in Tables 1 and 3 are averaged across seeds.

**Experimental Details for CS-Reward Learning CS-Reward with Multiple Tasks.** In the Multiple Tasks tasks experiment, the setup is similar to the multi-target experiment, but here we train the Alg. 1, on multiple tasks. The differences are:
In this experiment, the expert demonstrations $\mathcal{D}_{t_1}...\mathcal{D}_{t_T}$, are set to 10K demonstrations for each task. Discriminator updates per iteration $D_{\text{updates}}$ is set to 10. Number of tasks $T$ is set to 5 targets * 6 tasks. The generator updates per iteration $G_{\text{updates}}$, is set to 20.

**Experimental Details for Learning CS-Reward and Task-Reward with Multiple Tasks** Here, the experimental details are similar to experimental details for "Learning CS-Reward with Multiple Targets", but in this experiment we train the Alg. 2, that contains two discriminators, and the number of discriminator updates per iteration is the same for both discriminators.

**Compute Details:** Algorithm 1 can be run on a single NVIDIA T4 GPU. For improved time performance, parallelization is recommended.