# OpenReview forum: "Multi Task Inverse Reinforcement Learning for Common Sense Reward"
_ICLR.cc/2025/Conference — ICLR 2025 Conference Withdrawn Submission_

### Official Review · Reviewer_XGgV · 2024-10-29

**Soundness:** 1
**Presentation:** 3
**Contribution:** 2
**Rating:** 1
**Confidence:** 4

**Summary:**

The authors propose to do multi-task inverse reinforcement learning by decomposing the reward function into the sum of a task-specific reward, which they assume is given, and a common sense reward, which is learned and shared between the tasks.  Their proposed method uses AIRL to learn a discriminative common sense reward over multiple tasks, then transfers this common sense reward to a new task.  The authors also propose an extension of their method that does not require pre-specified task-specific reward by learning the task specific and common sense reward at the same time with two AIRL discriminators.  They conduct experiments on a subset of Meta-World tasks to demonstrate that their method requires multi-task training in order to learn an effective common sense reward and compares with other multi task IRL methods.

**Strengths:**

* Many of the ideas presented are interesting and worth exploring: disentangled reward functions, learning common sense rewards from multi-task data

**Weaknesses:**

* Their main method assumes task-specific rewards are given.  This is an unrealistic assumption and changes their problem setting from IRL to demonstration-guided RL.  In practice their method looks more like learning an auxiliary reward from multi-task demonstrations to supplement the task reward given by the environment.
* The unknown task reward extension of their method naively learns a task specific and common sense reward jointly without any effort at disentangling the two.  This will not automatically separate the reward functions and it’s unclear why this method would be better than learning a combined reward function.

There are also some serious issues with the experimental setup, which makes the paper’s main claims unconvincing to me.
* The authors only experiment on selected Meta-World tasks, which are all relatively similar tabletop manipulation tasks.  I am not convinced these results are generalizable to other benchmarks and types of tasks.
* Comparison methods.  While their method has access to both task specific rewards and expert demonstrations, the baselines methods seem to only have access to one or the other (reward for SAC, demos for IRL methods), so this is an unfair comparison.  Furthermore, they are missing other methods that decompose reward functions including Reward Network Distillation which they cite in related work.
* The definition of the target velocity and action norm common sense rewards seem contrived since they are not critical or even necessarily helpful to learning the tasks (since almost all experiments achieve 100% task success).  The common sense reward effectively learns a style preference from the demonstrations that’s not present in the task reward function.  The setup in section 5.4 where the algorithm extracts both task specific and common sense reward from demonstrations makes much more sense.
* Section 5.4 is missing any baseline methods, especially doing AIRL on each task individually, so I am not convinced that MT-CSIRL+LT is effectively decomposing the reward functions and transferring them.

**Questions:**

* For MT-CSIRL transferred onto an unseen task, do you  have access to that task’s task-specific reward and demonstrations?  Do you do any finetuning for the cs-reward?
* How are the MT-AIRL methods evaluated in Table 2?  Are they trained over all tasks’ demonstrations including the unseen task’s?  Do you use task-specific reward for MT-AIRL as well?

---

### Official Review · Reviewer_VYsK · 2024-10-31

**Soundness:** 1
**Presentation:** 3
**Contribution:** 1
**Rating:** 1
**Confidence:** 3

**Summary:**

The paper proposes to use multi-task experts to learn environmental constraints via a common sense reward. Then the authors demonstrate that by representing any task's reward function as common-sense reward+task-specific reward, agents can learn quickly and safely.

**Strengths:**

1. The paper addresses an important problem of learning a policy with learned environmental constraints using maxentIRL.

2.  The paper is easy to read and communicates the method clearly.

**Weaknesses:**

1. Motivation : The paper's motivation is based on unsubstantiated claims regarding capabilities and limitations of IRL. In line 76-82 the paper claims that "IRL fails to learn correct reward functions due to its connection to GANs". IRL is not equivalent to GAIL; rather GAIL is a way to solve distribution matching and obtained from a primal-dual perspective. While, methods like GAIL are not aiming to learn reward functions there exists method like f-IRL[2] and MaxEntIRL[3] that explicitly try to learn reward functions. f-IRL has looked at learning "stationary reward functions" - reward functions on which the policy can be retrained.
2. Literature review: I found the literature reviews to be somewhat shallow. For example, IRL was attributed to Arora and Doshi (2021); IRL existed long before and its origins can be traced back to [1] and possibly earlier. Authors are encouraged to use correct attributions and perform an in-depth literature review.
3. Unprincipled method: The reward function for any task is assumed to decompose as common sense reward+task-specific reward. Under such a decomposition there is no guarantee that optimizing task-specific reward results in same optimal policy as the combination with common sense reward.  Abeel and Ng[4] have proposed a class of reward functions that result in same optimal policy.
4: Assuming known structure of common sense rewards: In experiments it was presumed that common sense reward had a fix structure. I think that is a major limitation as the structure cannot be determined without domain knowledge.
5. Section 4.3 is titled curriculum learning, but I do not see the connection to curriculum learning from the method proposed.
6. Experiments: Standard IRL methods are missing from comparisons like MaxentIRL and f-IRL.


[1]: Abbeel, Pieter, and Andrew Y. Ng. "Apprenticeship learning via inverse reinforcement learning." Proceedings of the twenty-first international conference on Machine learning. 2004.
[2]: Ni, Tianwei, et al. "f-irl: Inverse reinforcement learning via state marginal matching." Conference on Robot Learning. PMLR, 2021.
[3]: Ziebart, Brian D., et al. "Maximum entropy inverse reinforcement learning." Aaai. Vol. 8. 2008.
[4]: Ng, Andrew Y., Daishi Harada, and Stuart Russell. "Policy invariance under reward transformations: Theory and application to reward shaping." Icml. Vol. 99. 1999.

**Questions:**

None

---

### Official Review · Reviewer_hmwd · 2024-11-03

**Soundness:** 2
**Presentation:** 3
**Contribution:** 2
**Rating:** 5
**Confidence:** 4

**Summary:**

This paper presents a framework that learns disentangled task-specific and task-shared common-sense rewards from multiple tasks using adversarial inverse reinforcement learning (IRL). The approach is simple and straightforward, based on the hypothesis that the reward has an additive structure. The method involves decomposing the reward term of traditional adversarial IRL methods. Results from robotic control experiments (Meta-World) indicate that learning from diverse tasks is crucial for capturing common-sense rewards, and the proposed method outperforms approaches that do not incorporate this common-sense reward learning.

Overall, this paper addresses an intriguing and significant challenge in reinforcement learning: how to learn useful common-sense rewards from demonstration data to facilitate generalization and adaptation. However, I think that certain technical aspects of the work, particularly the hypothesis regarding the reward structure, could be explored further to enhance its applicability to more general tasks and strengthen the framework. Therefore, I would assign a borderline rating (slightly negative), but I will consider adjusting this based on the authors' rebuttal and discussions with other reviewers.

**Strengths:**

- **[Motivation]**: This paper addresses a compelling and fundamental technical challenge in reinforcement learning: the need to learn common-sense rewards for improved generalization.

- **[Technical Soundness]**: The method is generally sound, with no major technical flaws. Its simplicity facilitates easy implementation and potential extension to other RL domains.

- **[Clarity of Presentation]**: The overall presentation is clear and easy to follow.

**Weaknesses:**

- **[Reward structure]**: The authors assume an additive reward structure, which may be overly simplistic. It would be beneficial to consider more complex functional forms that are representative of real-world scenarios, such as general non-linear functions. Additionally, incorporating noise factors in both common-sense and task-specific rewards—like additive noise, linear non-Gaussian models, or post-nonlinear models with noise—could enhance the framework. A discussion and analysis of these more general function forms, both theoretically (ensuring that disentanglement factors remain identifiable) and empirically (demonstrating that they can be learned within the framework), would be valuable.

- **[Common-sense reward concepts]**: The definition of common-sense rewards in this work is quite basic. Can the framework be scaled to learn more generalizable common-sense rewards, such as a foundational understanding of physical world models or other relevant norms applicable to more complex tasks, like those involving embodied agents? Incorporating language might also be a relevant avenue to explore [1]. This point is more of an open question, but any discussion around it would be appreciated.

- **[Relation to unsupervised RL]** In unsupervised RL, especially mutual-information-based skill discovery, methods can also learn a set of common skills from tasks. It might be beneficial to discuss that line of research and the major benefit of using this framework to learn disentangled common-sense rewards.

- **[Evaluation]**: Is there any theoretical or formal analysis addressing why common-sense reward learning performs poorly in single-task or single-target transfer scenarios compared to multi-task learning?

[1] Zhao, Zirui, Wee Sun Lee, and David Hsu. "Large language models as commonsense knowledge for large-scale task planning." Advances in Neural Information Processing Systems 36 (2024).

**Questions:**

I listed the questions in the above section together with weaknesses.

---

### Official Review · Reviewer_Cwin · 2024-11-04

**Soundness:** 2
**Presentation:** 1
**Contribution:** 2
**Rating:** 5
**Confidence:** 4

**Summary:**

This work proposes a method to recover a “common-sense” reward using demonstrations across multiple tasks. This reward represents the part of the reward function that may be common across tasks. The paper discusses why existing IRL methods like AIRL, GAIL fail to recover the true reward function upon convergence. For stable learning, a mechanism to scale the rewards using historical averages is proposed. Experiments on a few tasks in MetaWorld show that the proposed method can extract this reward and with the learned common-sense reward, the agent can learn new tasks with known (standard RL) or unknown task (IRL for task) rewards.

**Strengths:**

- The problem of recovering reward functions using demonstration is interesting as this will facilitate better transfer and generalization across tasks.
- The experiments on Meta-World show that the method can recover the reward functions that are common across tasks.

**Weaknesses:**

- The motivation is not clear. The point that an IRL algorithms (AIRL, GAIL) can fail to recover the true reward function is highlighted multiple times. However, it depends on the formalism as described in [1], where methods like GAIL is a primal approach and the learned reward at convergence might not align with the true reward, whereas solving the dual form of IRL where the reward function is learned with no-regret approach, the method should recover a meaningful reward function. So, this might explain that IRL methods when trained in Dual form can recover the true reward function and AIRL is not able to learn it.
- The paper is not well written and hard to follow, especially the experiments section.
- According to the paper, the IRL methods do not learn good reward functions. It is not clear why the proposed method still uses this formalism for the scenario with unknown task rewards (Section 4.4).

**Questions:**

1. IQ-Learn [2] showed that they could recover the reward function and should be added as a baseline for comparison.
2. When the task reward functions are known, then using Eq 6,7 with multiple tasks should recover the common reward function. However, for unknown task rewards (Section 4.4), Eq 8 will try to recover the task reward function. Given that Eq 8 uses the expert demonstration, will it recover the overall reward function?
3. If point 2 holds, a reward function trained with Eq 9 might not learn anything informative. This might be true because the learned task reward should learn about the sum of rewards from the expert trajectories. Here, I assume that the demonstration is obtained by experts trained with task rewards (Eq 10 /11).
3. At line 380, how are the trajectories sampled?
4. In Section 5.2, how will the performance be impacted if the agents cannot recover 100% success rate?
5. In Section 5.3, the method MT-CSIRL-LT uses a similar mechanism to IRL for learning the reward function and the policy. Since IRL does not recover good reward function, how well does the task reward correlate with the original task reward?
6. Overall, I feel the experiment section is hard to understand and needs to be explained better about the training and evaluation setups. The results presented in Table 1 are not clear.
7. As the episodes can terminate as soon as the task is solved, is it possible that the bias in rewards described in [3] leads to agents not following the constraints, and thereby the baselines in Table 2 do not learn desired behaviors?

#### References
[1] Swamy et al., Of Moments and Matching: A Game-Theoretic Framework for Closing the Imitation Gap, ICML'21\
[2] Garg et al., IQ-Learn: Inverse soft-Q Learning for Imitation, NeurIPS '21\
[3] Jena et al., Addressing reward bias in Adversarial Imitation Learning with neutral reward functions. Deep RL workshop at NeurIPS'2020.

---

### Note · Authors · 2024-11-24

I have read and agree with the venue's withdrawal policy on behalf of myself and my co-authors.